# Resistance Separation of Polymer Electrolyte Membrane Fuel Cell by Polarization Curve and Electrochemical Impedance Spectroscopy

**Jaehyeon Choi [1], Jaebong Sim [1] , Hwanyeong Oh [2] and Kyoungdoug Min [1],***

[1] Department of Mechanical and Aerospace Engineering, Seoul National University, 1 Gwanak-ro, Gwanak-Gu, Seoul 08826, Korea; wogus0922@naver.com (J.C.); sim6368@snu.ac.kr (J.S.)

[2] Fuel Cell Laboratory, Korea Institute of Energy Research, 152 Gajeong-ro, Yuseong-gu, Daejeon 34129, Korea; hyoh@kier.re.kr

* Correspondence: kdmin@snu.ac.kr; Tel.: +82-2-880-1661; Fax: +82-2-874-2001

**Abstract:** The separation of resistances during their measurement is important because it helps to identify contributors in polymer electrolyte membrane (PEM) fuel cell performance. The major methodologies for separating the resistances are electrochemical impedance spectroscopy (EIS) and polarization curves. In addition, an equivalent circuit was selected for EIS analysis. Although the equivalent circuit of PEM fuel cells has been extensively studied, less attention has been paid to the separation of resistances, including protonic resistance in the cathode catalyst layer (CCL). In this study, polarization curve and EIS analyses were conducted to separate resistances considering the charge transfer resistance, mass transport resistance, high frequency resistance, and protonic resistance in the CCL. A general solution was mathematically derived using the recursion formula. Consequently, resistances were separated and analyzed with respect to variations in relative humidity in the entire current density region. In the case of ohmic resistance, high frequency resistance was almost constant in the main operating load range (0.038–0.050 $\Omega$ cm$^2$), while protonic resistance in the CCL exhibited sensitivity (0.025–0.082 $\Omega$ cm$^2$) owing to oxygen diffusion and water content.

**Keywords:** resistance separation; overpotential; electrochemical impedance spectroscopy; polarization curve; protonic resistance

## 1. Introduction

Owing to increasing greenhouse gas emissions, global warming has become one of the biggest environmental problems of the 21st century. Approximately 17% of the $CO_2$ emissions originate from automotive internal combustion engines [1,2]. Accordingly, automobiles with polymer electrolyte membrane (PEM) fuel cells have gained attention, with the aim of achieving zero emissions and higher efficiency [3]. A PEM fuel cell converts the chemical energy of the reaction between $H_2$ and $O_2$ into electrical energy using pure water [3,4]. Hence, PEM fuel cell technologies have received significant research interest.

In this process, researchers require useful data on water management, ohmic losses, and the ionic conductivity of PEM to identify the research direction and operating conditions. The most frequently used methods are electrochemical impedance spectroscopy (EIS) and polarization curves. These have been generally used to procure information on resistance and current voltage, respectively [5,6].

As the technique for determining the equivalent circuit of the EIS technique is not proprietary [7], it has been the subject of study for a long time. It has been widely used in the case of lumped circuits, for which many types of equivalent circuits have been suggested [4,8–10]. However, the lumped circuit is not appropriate to describe the protonic resistance in the cathode catalyst layer (CCL), which represents a 45-degree straight line in the high-frequency region around the first arc [11–14]. It pushes away the first semi-

circle from the origin, and the distance from the x-intercept is proportional to the protonic resistance in the CCL. This distance is called the effective protonic resistance [15–20].

To describe this line, differential equations or the finite transmission-line model (TLM) are normally used [13,21,22]. Makharia et al. [13] reported the calculation of effective protonic resistance using differential equations and analyzed the results of the $H_2/N_2$ experiment. The sum of all the distributed proton transport resistances estimated from the TLM was approximately equal to three times the effective protonic resistance. However, in this process, the homogeneity of distributed elements in the TLM was assumed, and the experiments were carried out at low current densities.

Malevich et al. [23] studied the effect of inhomogeneity of elements in the TLM in the $H_2/N_2$ condition. The results showed that non-uniform distribution of proton conductivity and capacity affected the shape and angle of the 45-degree straight line around the high-frequency region. Thus, the effective protonic resistance in CCL was not one-third of the protonic resistance in CCL. Furthermore, Gerteisen [24] studied the impact of the inhomogeneity of charge transfer resistance in the $H_2/air$ condition, which affected the charge transfer arc.

As the non-uniformity of charge transfer resistance affected the total resistance, Gaumont et al. [16] suggested a general solution that included a non-uniform distributed charge transfer resistance in the CCL. The general solution was derived using an equivalent circuit with current distribution throughout the thickness of the electrode. Although this work considered the variation in distributed charge transfer resistance, the equivalent circuit did not take into account the effects of mass transport resistance or diffusion.

Cruz-Manzo and Chen [1] suggested using the TLM together with the Bounded Warburg to reflect the diffusion effect in the CCL in the entire current density region [25]. The authors evaluated the differential equations and suggested an electrical equivalent circuit but did not attempt an estimation using a recursion formula. To solve the differential equations, it was assumed that the protonic resistance in CCL ought to be much smaller than the charge transfer resistance.

While EIS analysis provides detailed information on each component of resistance, the polarization curve is also commonly used to investigate the general quantification of performance [5]. As each of these two experiments provides different sets of information, several researchers have used both [4,26,27]. Tang et al. [28] compared the cell voltage drop owing to each resistance component using EIS and polarization curves. To compare the two experimental results, each resistance component was evaluated via EIS, following which the voltage drop was estimated and compared with that obtained from the polarization curve. Consequently, the authors suggested a methodology to compare the results of EIS with those of the polarization curve; however, the protonic resistance in the CCL was not considered.

Several works have been conducted using EIS analysis and polarization curves. However, owing to deficiencies in the methodologies' ability to separate the estimation of each resistance, the estimation of each component of resistance (charge transfer resistance, high frequency resistance (*HFR*), mass transport resistance, and effective protonic resistance in the CCL) has not been sufficiently carried out. In the present study, these four types of resistances, especially including effective protonic resistance in the CCL, were considered to derive a general solution of the impedance model. From this general solution, a new method was proposed to evaluate the total resistance.

When the distributed elements in the TLM were homogeneous and the distributed protonic resistance in the CCL was less than the summation of the distributed charge transfer resistance and mass transport resistance, each resistance component was evaluated using EIS alone. In other cases, the correlation between EIS analysis and the polarization curve was used. Thus, the total resistance of a PEM fuel cell was quantitatively determined. This was conducted by varying current densities and conditions of relative humidity (RH), EDIT, and the resultant variations in resistance were analyzed.

## 2. Experiment

### 2.1. Experimental Setup

The measurements for EIS analysis and polarization curve were carried out under the experimental conditions listed in Table 1. A commercialized product called GORE$^{TM}$ PRIMEA$^®$ 5730 was used to improve the validity and reproducibility of the experimental results. Before measuring the EIS data, the voltage and current density were measured to plot the polarization curve, because the measurements in both cases were carried out under identical conditions. For these experiments, an FC impedance meter (KFM 2150 and PLZ-4W) was used. To satisfy EIS assumptions such as linearity and stability, the potential amplitude was maintained at lower than 10 mV. For the sinusoidal alternating current (AC) signal, the frequency range analyzed was 20 kHz to 900 mHz. However, to ensure the existence of another semi-circle appearing at low frequency, the end frequency was 10 mHz in the high current density region. The high frequency resistance was measured at 20 kHz resistance. The electronic bulk resistance and contact resistance components of the *HFR* were measured using an ex situ direct current (DC) experiment with a PEM fuel cell, except for the membrane electrode assembly (MEA). The temperature and pressure were maintained at 65 °C and ambient pressure, respectively.

**Table 1.** Operating conditions for measuring the EIS and polarization curve.

| Parameter | Condition |
| --- | --- |
| Test mode | Galvanostatic technique |
| Frequency | 20 kHz to 900 mHz |
| Swing width of AC current | within a voltage amplitude of less than 10 mV |
| Current density | 0.1–2.4 A/cm$^2$ |
| Mass flow | Anode: 0.400 ln/min (SR * > 20)<br>Cathode: 2.00 ln/min (SR * > 40) |
| Reactant gas | H$_2$/air<br>H$_2$/Heliox (21% O$_2$ with the balanced made of He) |
| Inlet gas RH ** | 50, 80, 100% (anode/cathode) |
| Cell temperature | 65 °C |
| Outlet pressure | Ambient pressure |

* Stoichiometric ratio, ** relative humidity.

Table 2 lists the materials of the fuel cell used in this work. As shown in Figure 1, the 1 cm$^2$ active area near the outlet of the 25 cm$^2$ flow channel was assembled using a gasket and gas diffusion layer (GDL), which provides in-plane uniformity to resistance, current distribution, and reactant gas. Thus, a single cell was used to control the operating conditions as well.

**Table 2.** Specifications of the unit cell.

| Component | Condition |
| --- | --- |
| Flow Channel | Parallel channels (anode/cathode)<br>1/0.815 mm width (channel/rib)<br>0.4/0.6 mm depth (anode/cathode) |
| GDL-MPL | JNT30-A6H<br>(Thickness 325 ± 5 μm) |
| MEA $^†$ | GORE$^{TM}$ PRIMEA$^®$ 5730 $^‡$ |

$^†$ Membrane electrode assembly, $^‡$ catalyst-coated membrane.

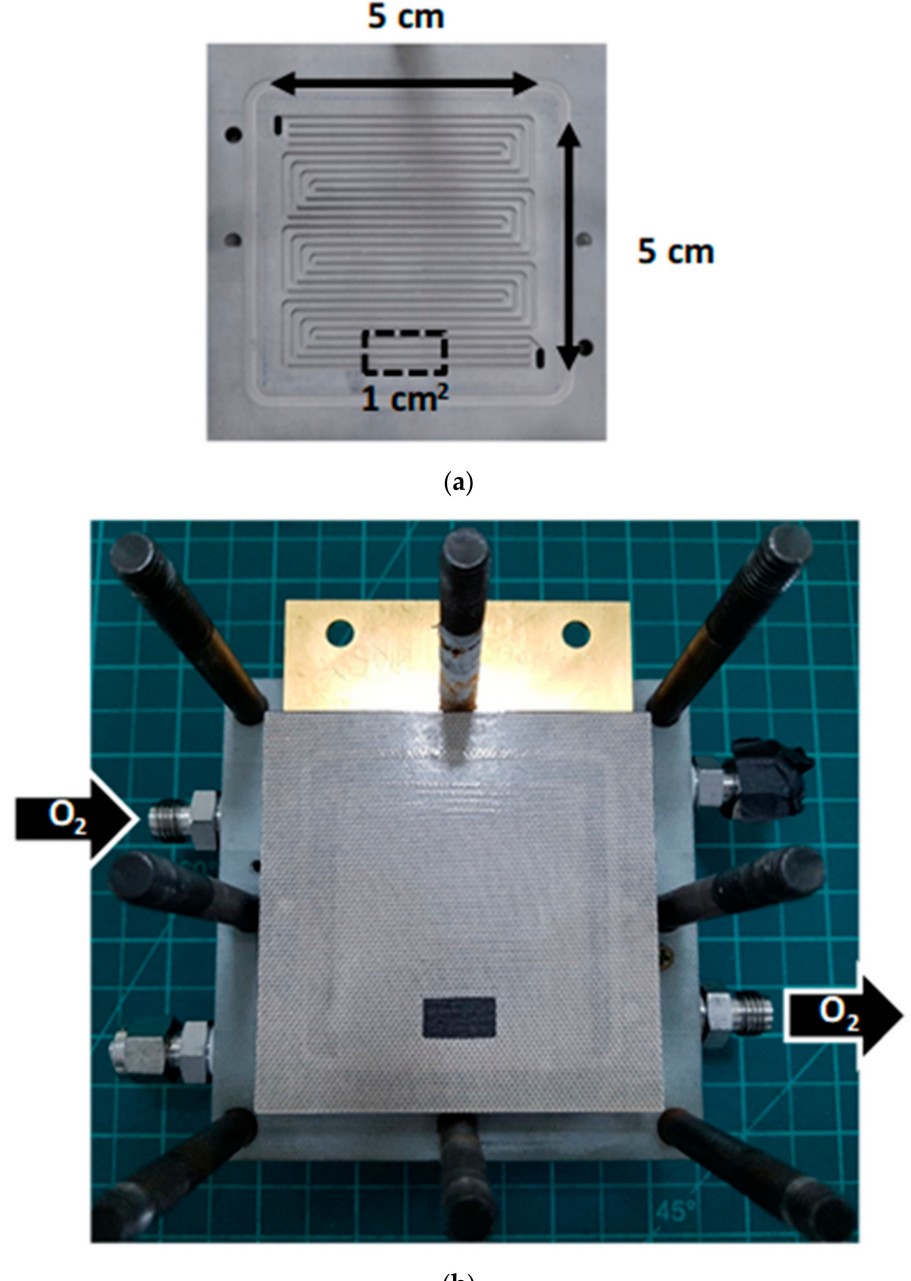

**Figure 1.** Setup of the active area. (**a**) Flow channel. (**b**) Assembly of end plate, flow channel, gasket, and gas diffusion layer (GDL).

### 2.2. Experimental Conditions and Assumptions

Unlike in other fuel cell experiments, the use of EIS analysis and polarization curves in the present work requires high stoichiometry ratios (SR) and a small active area for the following assumptions related to the impedance model.

- The SR has to be kept high to minimize reactant depletion along the channels. This setup also creates a condition of high diffusion through the CCL.
- The active area of the fuel cell needs to be small to have a uniform pressure and velocity.

This study used the current distribution equation, Equation (7), suggested by Eikerling and Kornyshev [29], which assumed fast oxygen diffusion, and hence a high SR condition was applied. The RH of the inlet gas was varied in the range from 50% to 100% to investigate

its effect on resistance. The current was varied in the range from 0.1 to 2.4 A/cm$^2$ in increments of 0.1 A/cm$^2$ to compare the EIS and polarization curve results.

### 3. Impedance Model

*3.1. Electrical Equivalent Circuit*

A reasonable selection of an electrical equivalent circuit is an essential step before the analysis of the experimental data. The rate of hydrogen oxidation is greater than that of oxygen reduction reaction, and hence, the anode resistance can be neglected [17,30–32]. In addition, the ohmic resistance caused by the electron flow in the CCL can be neglected because the electronic resistance in the CCL is smaller than the ionic resistance. Moreover, the anode resistance and the electronic resistance in the CCL were not critical, as shown in our experimental results, and were therefore neglected as in previous studies [13,16,17,30–32]. Unlike other equivalent circuits, the bounded constant phase element (BCPE) in the TLM was required to represent the mass transport effect in the CCL. Thus, the equivalent circuit suggested by Cruz-Manzo and Chen [1] was used in the present work. To improve the fitting results, the Bounded Warburg was replaced with the BCPE, and the element of the inductor was eliminated, as shown in Figure 2.

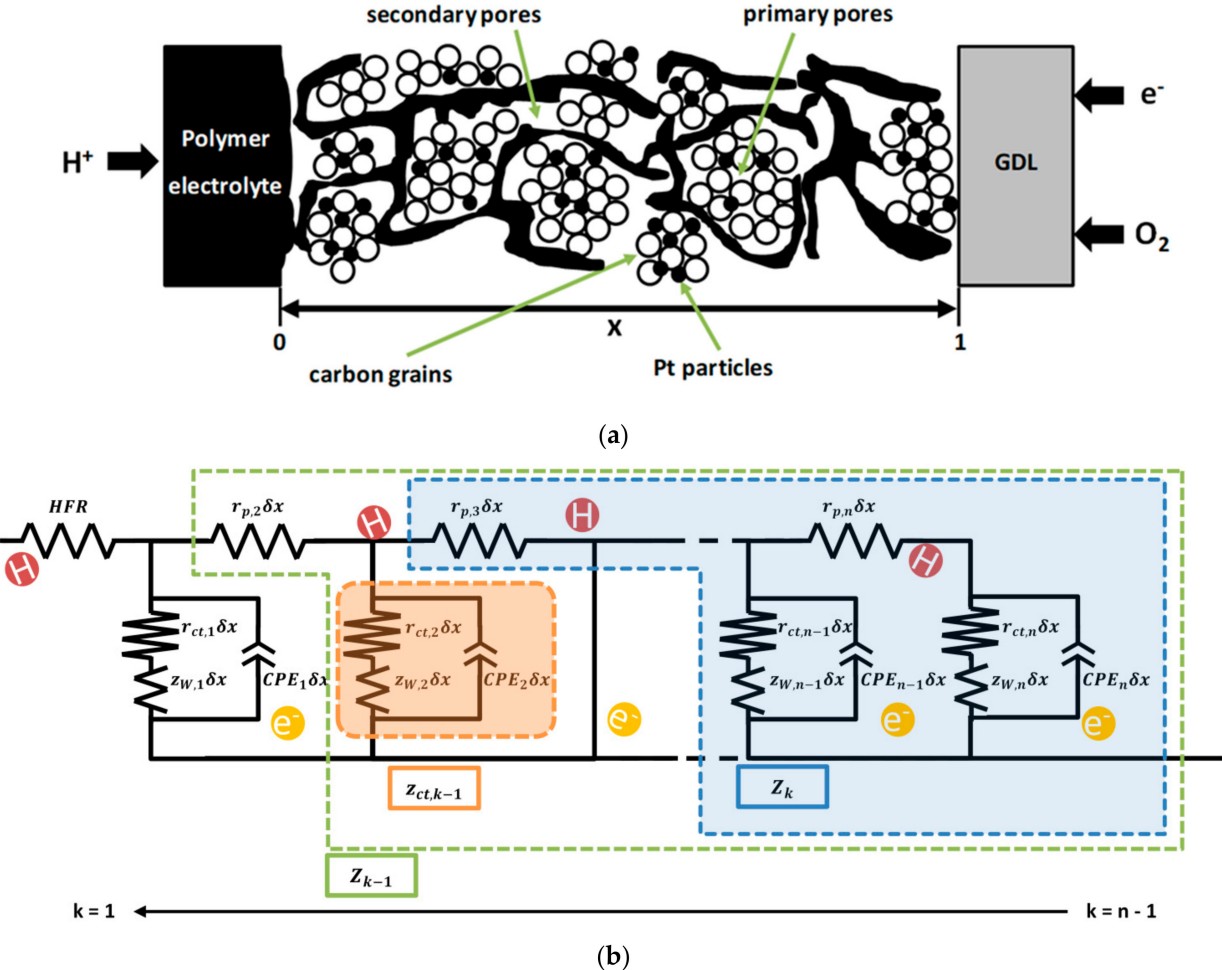

**Figure 2.** (**a**) A schematic of the catalyst layer (CCL). $x$ represents the non-dimensional thickness of the CCL from the CCL/GDL interface ($x = 1$) to the polymer electrolyte membrane (PEM)/CCL interface ($x = 0$). (**b**) Electrical equivalent circuit of a PEM fuel cell represented by a transmission-line model (TLM) [1].

The *HFR* includes the electronic bulk resistance, contact resistance, and resistance due to proton conduction in the membrane. The TLM shown in Figure 2 represents the catalyst

layer, which includes the distributed protonic resistance in the CCL ($r_p$), distributed charge transfer resistance ($r_{ct}$), distributed BCPE ($z_W$), and constant phase element (CPE) for the double-layer capacitance. Here, $n$ is the total number of iterations, and $k$ is the iteration number of the node in the interval ($1 \leq k \leq n-1$). Referring to a previous work, $n$ is 1000 [16].

### 3.2. General Solution

To represent the Nyquist plot using the TLM, the general solution was used with the recursion formula [16]. The CCL section of the Nyquist plot can be evaluated iteratively using the recursion formula from the CCL/GDL interface to the PEM/CCL interface as follows.

$$\begin{cases} Z_n = z_{ct,n} + r_{p,n}\delta x \\ Z_k = \left(\frac{1}{Z_{k+1}} + \frac{1}{z_{ct,k}}\right)^{-1} + r_{p,k}\delta x \,, \, r_{p,1} = 0 \end{cases} \tag{1}$$

with

$$\begin{cases} \delta x = \frac{l}{n} \\ CPE_k = q_{C,k}(iw)^{P_{C,k}} \\ z_{ct,\,k} = \left[\left\{r_{ct,k}\delta x + z_{W,k}\delta x\right\}^{-1} + CPE_k\delta x\right]^{-1} \\ z_{W,k} = \frac{tanh\left(r_{mass,k}\delta x^2 \, q_{B,k}(iw)^{P_{B,k}}\right)}{q_{B,k}(iw)^{P_{B,k}} \, \delta x^2} \end{cases} \tag{2}$$

where $l$, $x$, $r_{mass}$, $q_C$, $P_C$, $i$, $w$, $q_B$ and $P_B$ are the thickness of the CCL, the non-dimensional distance along the CCL ($0 \leq x \leq 1$), the distributed mass transport resistance, the distributed parameter related to CPE, the CPE exponent, the imaginary component in the impedance, the frequency, the distributed parameter related to BCPE, and the BCPE exponent, respectively.

To apply the inhomogeneity of $r_{ct}$, $r_{ct}$ is calculated as follows [16,29]

$$r_{ct,k} = \frac{b}{j(x)} \frac{1}{\delta x^2} \tag{3}$$

with

$$j(x) = \frac{\sqrt{2i^*\sigma b}}{l} \cdot \exp\left(\frac{\eta_1}{2b}\right) \cdot tan\left(\sqrt{\frac{i^*}{2\sigma b}}(1-x)\exp\left(\frac{\eta_1}{2b}\right)\right) \tag{4}$$

where $b$ is the Tafel slope, $\sigma$ is the proton conductivity, $i^*$ is the exchange current density, and $\eta_1$ is the overpotential at the PEM/CCL interface. This equation can be used under the assumption of rapid oxygen diffusion. A single parameter, Equation (5), is used to check this assumption in this experimental condition. The detailed derivation of the Equations (4) and (5) is given in Ref. [29].

$$g = \frac{4FPD^{eff}}{RTl} \frac{1}{\sigma b} \tag{5}$$

It was estimated to be in the range 14–250 based on the experimental results. $D^{eff}$ and $\sigma$ can be estimated from BCPE. Thus, Equation (4) can be used in this study because $g \gg 1$.

### 3.3. Resistance Separation Using EIS

The general solution with the recursion formula was derived in the last section. However, the distributed resistance from the general solution is not convenient to separate the total resistance ($R_{total}$). As the experimental conditions, two techniques were suggested. First, when the distributed elements are homogeneous and $r_p/(r_{ct} + r_{mass}) \ll 1$, resistance separation is conducted using EIS. On the other hand, not only EIS but also the polarization curve are conducted when the distributed elements are not homogeneous or $r_p/(r_{ct} + r_{mass})$ is not much smaller than 1. In this section, the first case was investigated. The second case was investigated in Section 3.5. To divide $R_{total}$ into the effective protonic resistance in the

CCL ($R_p^{eff}$), charge transfer resistance ($R_{ct}$), and mass transport resistance ($R_{mass}$) using the distributed elements, three assumptions were applied.

- The distributed elements in the CCL are homogeneous [33].
- $r_p/(r_{ct} + r_{mass}) \ll 1$.
- As previously mentioned, the anode resistance and the electronic resistance in the CCL are neglected.

To check the validity of the first assumption, a Nyquist plot was plotted using Equation (1) with homogeneous distributed elements, and then it was compared to the EIS experimental results.

$R_{total}$ is the most important information obtained in this study. A point at the intersection of the real axis at a low frequency is extracted from Equation (1) [34].

$$R_{total} = \lim_{w \to 0} (HFR + Z_1)$$
$$= HFR + \frac{r_{ct}\delta x + r_{mass}\delta x}{A_{n-1} + 1} + r_p \delta x \tag{6}$$

with

$$A_{n-k} = \frac{(A_{n-k-1}+1)(r_{ct}+r_{mass})}{(r_{ct}+r_{mass})+(A_{n-k-1}+1)r_p}, \quad A_0 = 0 \tag{7}$$

Then, a general term can be derived as follows. From Equation (7),

$$A_{n-k} = A_{n-k-1} + 1 - \frac{r_p}{r_{ct}+r_{mass}} \cdot \frac{(A_{n-k-1}+1)^2}{1 + (A_{n-k-1}+1) \cdot \frac{r_p}{r_{ct}+r_{mass}}} \tag{8}$$

Based on the second assumption,

$$A_{n-k} \approx A_{n-k-1} + 1 - \frac{r_p}{r_{ct}+r_{mass}} \frac{(A_{n-k-1}+1)^2}{1} \tag{9}$$

When k is large, Equation (8) can be approximated to a simple form as shown below; because both $r_p/(r_{ct}+r_{mass})$ and $(A_{n-k-1}+1)^2$ are small.

$$A_{n-k} \approx A_{n-k-1} + 1 \tag{10}$$

$$A_{n-k} \approx n - k \tag{11}$$

Therefore, the right-hand side of Equation (9) can be simplified as Equation (12) using Equation (11). At this point, the error caused by Equation (11) is only acceptable when k is large. Therefore, the error in the third term on the right-hand side can be approximated using Equation (11) due to $r_p/(r_{ct}+r_{mass})$. However, Equation (11) is not applicable to the first term on the right-hand side, because the error in this term is not negligible without multiplication by $r_p/(r_{ct}+r_{mass})$.

$$A_{n-k} \approx A_{n-k-1} + 1 - \frac{r_p}{r_{ct}+r_{mass}} \cdot (n-k)^2 \tag{12}$$

Finally, the general term of the recursion formula is derived from Equation (12).

$$A_{n-k} \approx n - k - \frac{r_p}{r_{ct}+r_{mass}} \frac{(n-k)(n-k+1)(2n-2k+1)}{6} \tag{13}$$

To check the derivation of this equation, Equations (7), (11), and (13) are plotted as shown in Figure 3. When k is large, three lines are well matched. However, Equation (11) is not well matched when k is small.

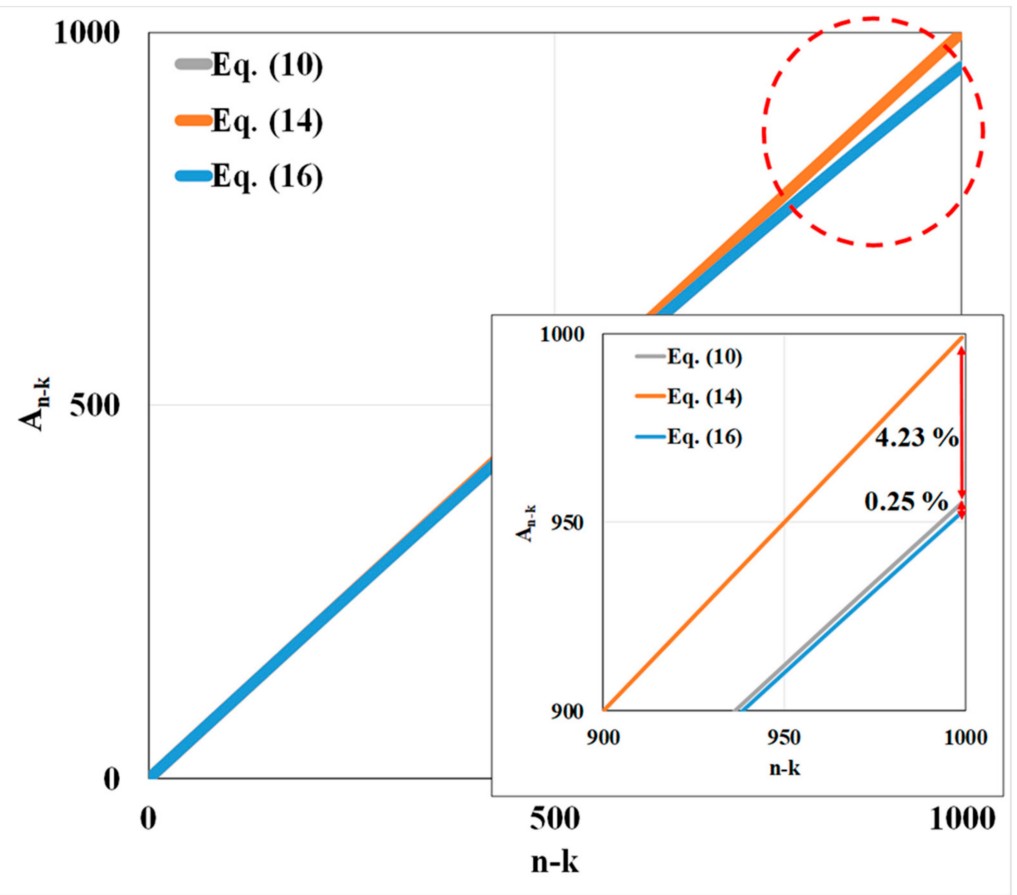

**Figure 3.** Equation (7) is simplified. Equation (11) can be used only if the k is large. On the other hand, the error in Equation (13) is less than 0.25% at 0.1 A/cm$^2$, so this equation can be used regardless of the k.

The general term can be derived by substituting Equation (13) into Equation (6).

$$R_{total} = HFR \quad + \frac{r_{ct}\delta x + r_{mass}\delta x}{n-1-\frac{r_p}{r_{ct}+r_{mass}}\frac{(n-1)(n)(2n-1)}{6}+1} + r_p\delta x \tag{14}$$

When the first assumption is valid [33], Equation (14) can be rearranged as follows [17,34].

$$\begin{aligned} R_{total} &= R_{ct} + R_{ohmic} + R_{mass} \\ &= R_{ct} + HFR + R_P^{eff} + R_{mass} \end{aligned} \tag{15}$$

with

$$R_P^{eff} = \frac{\frac{r_p\delta x(n-1)(2n-1)}{6n}}{1-\frac{r_p}{r_{ct}+r_{mass}}\frac{(n-1)(2n-1)}{6}} + r_p\delta x \tag{16}$$

$$R_{ct} = \frac{r_{ct}\delta x}{n} \tag{17}$$

$$R_{mass} = \frac{r_{mass}\delta x}{n} \tag{18}$$

where $R_{ohmic}$ is the ohmic resistance of the summation of $HFR$ and $R_P^{eff}$.

### 3.4. Validation of Recursion Formula

In this section, the general solution (Equations (1)–(4)) and the general term (obtained by substituting Equation (13) into Equation (6)) were validated using the solution suggested by Cruz-Manzo and Chen [1] and our experimental results.

Cruz-Manzo and Chen [1] suggested not only an equivalent circuit but also a solution of differential equations for the impedance model depicted in Figure 2. This solution, the validation model, was used to validate the general solution. After disregarding the inductance term and replacing the Bounded Warburg with the BCPE, the final total impedance equation can be written as follows:

$$Z_{total} = HFR + \frac{[R_{ct} + Z_W]\gamma_1 \coth(\gamma_1(1-x))}{1 + Q_C(iw)^{P_C}[R_{ct} + Z_W]} \tag{19}$$

with $\gamma_1 = \sqrt{R_p\left[\frac{1}{R_{ct}+Z_W} + Q_C(iw)^{P_C}\right]}$

$$Z_W = \frac{\tanh\left(R_{mass}Q_B(iw)^{P_B}\right)}{Q_B(iw)^{P_B}} \tag{20}$$

where $R_p$ is protonic resistance in the CCL, $Z_W$ is the BCPE, $Q_C$ is the parameter related to the CPE, and $Q_B$ is the parameter related to the BCPE. To derive Equation (19), it was assumed that the element in the CCL was homogeneous. Furthermore, to satisfy the homogeneity of $r_{ct}$, Equation (21) should be valid [1].

$$\frac{R_p}{R_{ct}} \ll 1 \tag{21}$$

Figure 4 shows the fitting results of the general solution and validation model. Both models fitted well with the experimental results at RH 80% and 100%, regardless of the current densities. However, the general solution did not fit well with the validation model at RH 50%. The results can be explained based on the assumption of the homogeneity of $r_{ct}$. The value of $r_p/(r_{ct} + r_{mass})$ was much smaller than 1 ($3.5 \times 10^{-7} > r_p/(r_{ct} + r_{mass}) > 7.6 \times 10^{-8}$) regardless of the RH conditions. Thus, it was inferred that the mismatch at RH 50% was caused by the inhomogeneity of $r_{ct}$ in the TLM. As a result, the homogeneous assumption of $r_{ct}$ can be valid at RH 80% and 100%.

On the other hand, as the straight line in our experimental data was maintained at 45 degrees, it seems that the impacts of the inhomogeneity of the double layer capacity and protonic conductivity might not be significant, similar to what was found in previous research [24].

Furthermore, the general term was validated using $R_{total}$, which was evaluated from the experimental data. By comparing the $R_{total}$ values obtained from the general term, Equation (18), and the general solution, Equation (9), the error of the general term was less than 1% at RH 100%, regardless of the current density. However, the error increased as RH decreased. The maximum error at RH 80% was around 1.5%. Therefore, the error caused by the assumption that $r_p/(r_{ct} + r_{mass}) \ll 1$ was not significant at RH 80%, 100%. Consequently, resistance separation can be carried out using Equations (15)–(18) when the TLM is homogeneous and RH is greater than 80%.

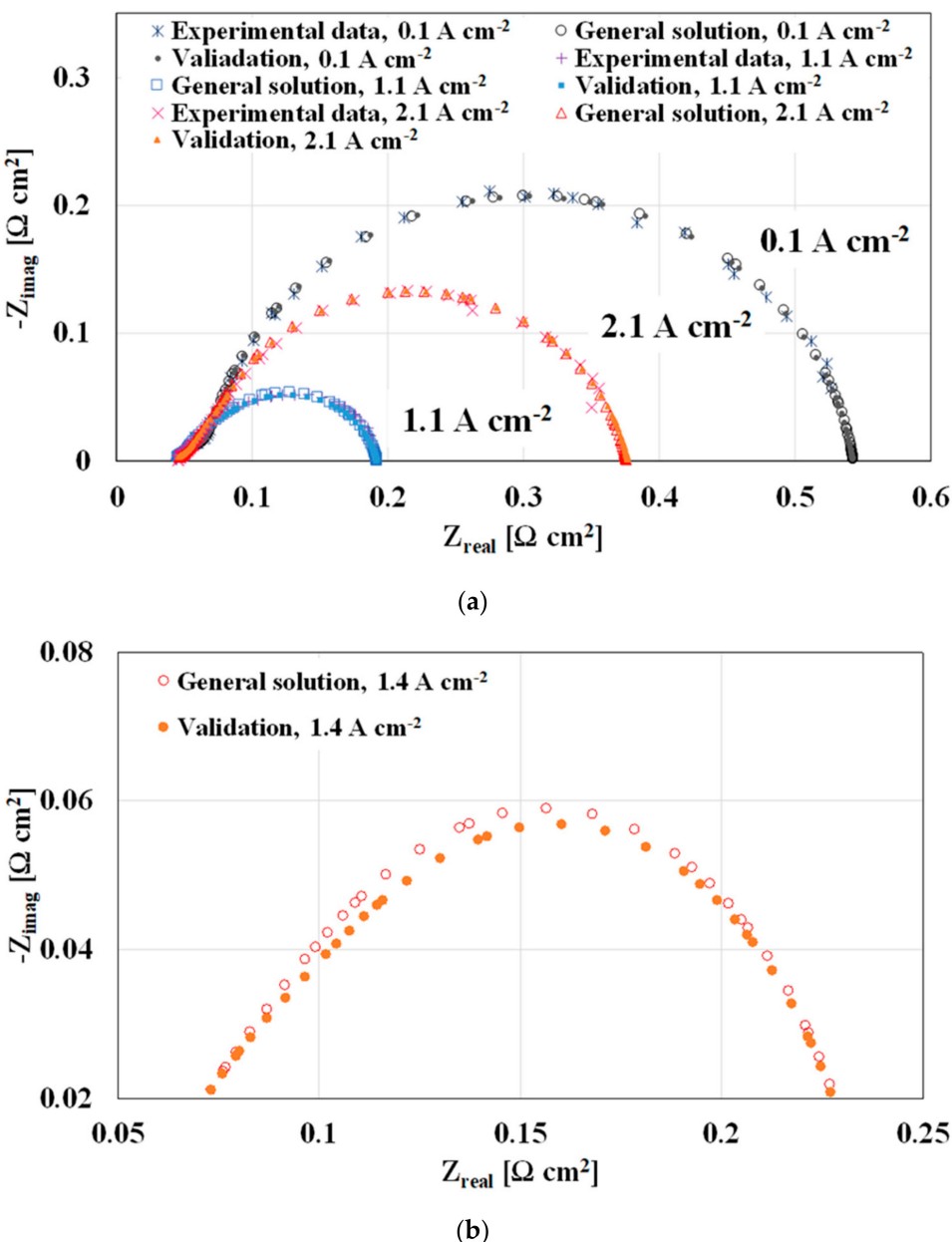

**Figure 4.** (**a**) Nyquist plots are plotted at relative humidity (RH) 100%, (**b**) RH 50% under $H_2$/air condition. Open symbols represent general solution and closed symbols represent validation model. Cross symbols represent experimental data.

In addition, $R_P^{eff} \approx R_p/3$ can be derived. From Equation (19), because $n \gg 1$ (in this study, $n$ is 1000), $n \approx n - 1 \approx \frac{2n-1}{2}$.

$$R_P^{eff} \approx \frac{\frac{nr_p\delta x}{3}}{1 - \frac{r_p}{r_{ct}+r_{mass}}\frac{n^2}{3}} + r_p\delta x \tag{22}$$

If $r_p\delta x^2/(r_{ct} + r_{mass}) \approx 0$ and $r_p\delta x \ll \frac{nr_p\delta x}{3}$,

$$R_P^{eff} \approx \frac{r_p\delta x\left(1 - \frac{r_p}{r_{ct}+r_{mass}}\frac{n^2}{3}\right) + \frac{nr_p\delta x}{3}}{1 - \frac{r_p}{r_{ct}+r_{mass}}\frac{n^2}{3}}$$
$$\approx \frac{nr_p\delta x}{3}$$
$$\approx \frac{R_p}{3} \tag{23}$$

### 3.5. Resistance Separation Using the Correlation between EIS and Polarization Curve

In the last two sections, resistance separation using EIS was investigated, when the distributed elements are homogeneous and approximation of $r_p/(r_{ct} + r_{mass})$ is valid. However, in some cases, the approximation of $r_p/(r_{ct} + r_{mass})$ or the assumption onf the homogeneity of $r_{ct}$ may not be valid. The experimental results of middle or high current densities under the $H_2$/Heliox condition showed that the assumption $r_p/(r_{ct} + r_{mass}) \ll 1$ was not correct, because $r_{mass}$ was low. This section suggests another methodology for a use under this condition, because the resistance separation using Equations (15)–(18) is no longer applicable. For this reason, the correlation between EIS and polarization curve is applied to separate the resistance. Naturally, resistance separation can be carried out using a polarization curve alone [28]; however, the additional analysis of the EIS results is helpful to consolidate the validity of the resistance separation.

The polarization curves, shown in Figure 5, are expressed using a semi-empirical equation [35]:

$$E = E_0 - b \ln\left(I \cdot 10^3\right) - IR_{ohmic} - m_{mass} \exp(n_{mass} I) \tag{24}$$

where $E$ is the cell voltage, $E_0$ is the open-circuit voltage (OCV), $I$ is the current density, $m_{mass}$ is the mass transport coefficient, and $n_{mass}$ is the simulation parameter for the polarization curve fitting. For resistance separation based on Equation (24) [17],

$$
\begin{aligned}
R_{total} &= R_{ct} + R_{ohmic} + R_{mass} \\
&= \frac{dE}{dI} \\
&= \frac{b}{I} + R_{ohmic} + m_{mass} n_{mass} \exp(n_{mass} I) \\
&= \frac{b}{I} + HFR + R_P^{eff} + m_{mass} n_{mass} \exp(n_{mass} I)
\end{aligned}
\tag{25}
$$

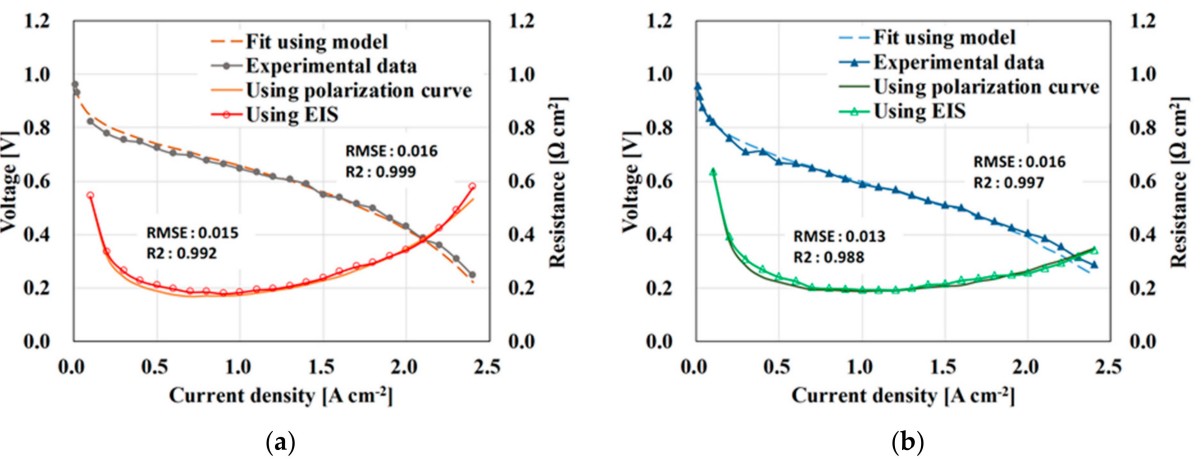

**Figure 5.** Polarization curves and total resistances at (**a**) RH 100%, (**b**) RH 80% under $H_2$/air experiments. The experimental data of polarization curves were fitted using semi-empirical model. The total resistances using electrochemical impedance spectroscopy (EIS) were matched with the total resistances using polarization curves.

*HFR* is evaluated via EIS regardless of the current density region. Meanwhile, $R_{ct}$ and $R_{mass}$ are estimated using EIS analysis and the polarization curve. In general, in the region of low current density, the approximation of $r_p/(r_{ct} + r_{mass})$ and the assumption of the homogeneity of $r_{ct}$ can be used [13,16]. Thus, the unknown parameters in Equation (24), namely $E_0$, $b$, $m_{mass}$, and $n_{mass}$, are determined using the experimental data of the EIS at low current densities. By Equation (25), the values of $R_{ct}$ and $R_{mass}$ can be estimated when $r_p/(r_{ct} + r_{mass})$ is large or $r_{ct}$ is not homogeneous. Finally, $R_P^{eff}$ can be evaluated using Equation (25), because $R_{total}$ is found by the EIS and polarization curve. Consequently, although $r_p/(r_{ct} + r_{mass})$ is not small and $r_{ct}$ is not homogeneous, the resistance can be separated using Equation (25) and the fitting results of the other current density regions.

Summary of solutions explain on Table 3.

**Table 3.** Summary of solutions.

|  | $R_p$ and $R_{mass}$ | Inhomogeneity of Elements | Resistance Separation |
|---|---|---|---|
| Solution suggested by Cruz-Manzo and Chen | Considered | Not considered | Facile |
| General solution | Considered | Considered | Arduousness |
| Approximated solution with homogeneous elements | Considered | Not considered | Facile |
| Approximated solution with inhomogeneous elements | Considered | Considered | Facile |

## 4. Results and Discussion

### 4.1. Comparison between EIS and Polarization Curve

As shown in Figure 5, the polarization curves obtained from our experimental results fitted well with fitting Equation (24). Differentiating the fitting equation with respect to *I* provided the information on resistance, as mentioned in Section 3. These results were also fitted well with the experimental data of the EIS, as shown in Figure 6 ($R^2 > 0.995$ and RMSE < 0.01). As reported in previous studies [36,37], $R_{ct}$ increased as RH decreased. The $R_{mass}$ at RH 100% was greater than that at RH 80% in the high current density region; however, these results were not maintained in the low current density region [4,38,39]. Furthermore, from the results of the $R_{ohmic}$ at low current densities, as shown in Figure 7, the generated water in the CCL also affected $R_{ct}$ and $R_{mass}$ owing to the change in the ORR pathways, proton activity, and catalyst surface condition [36].

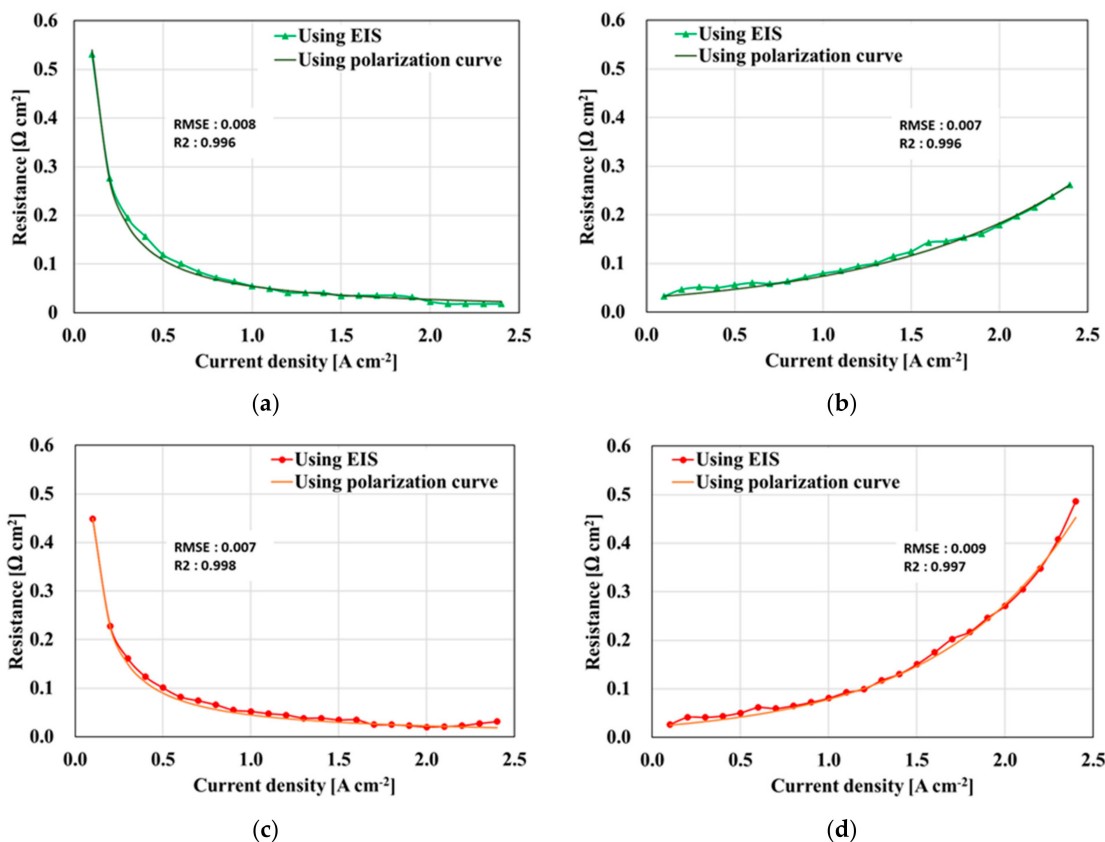

**Figure 6.** Results of the resistance separation under $H_2$/air. (**a**) Charge transfer resistance at RH 80%. (**b**) Mass transport resistance at RH 80%. (**c**) Charge transfer resistance at RH 100%. (**d**) Mass transport resistance at RH 100%.

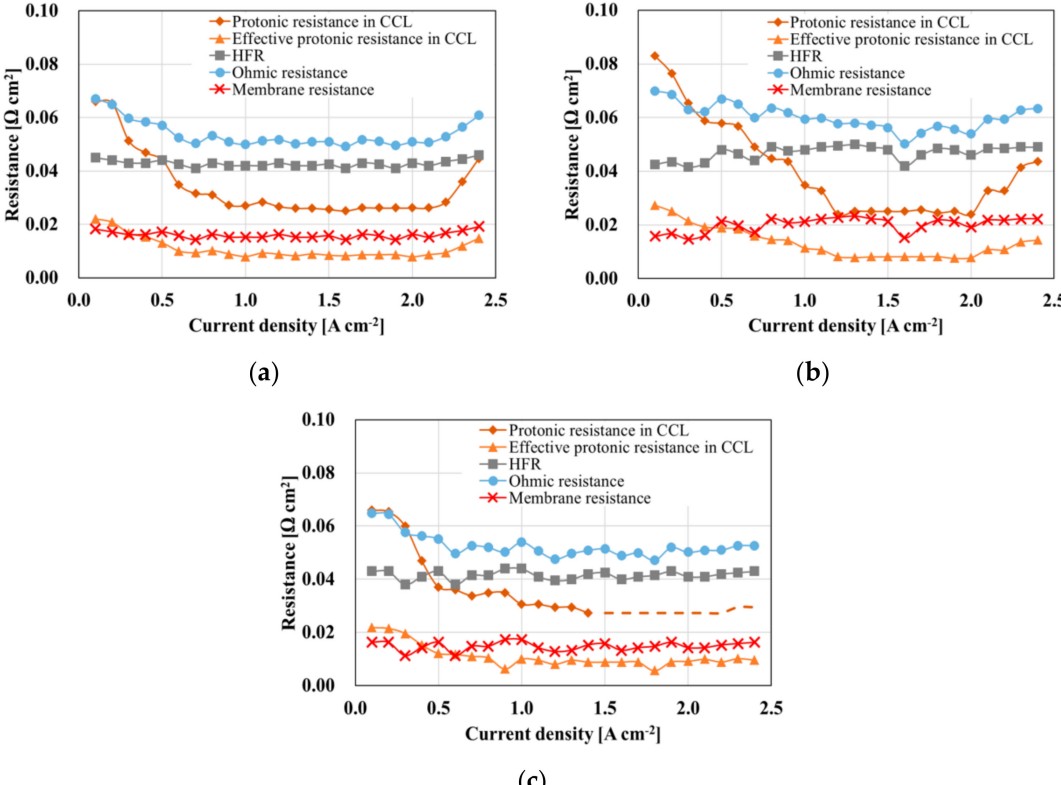

**Figure 7.** Separation of the ohmic resistance at (**a**) RH 100% under $H_2$/air, (**b**) RH 80% under $H_2$/air, and (**c**) RH 100% under $H_2$/Heliox. The rhombus symbol is the $R_p$, triangle symbol is the $R_P^{eff}$, the square symbol is *HFR*, and the circle symbol is ohmic resistance. At the dotted line, the fitting of the EIS was not well fitted, so protonic resistance in CCL cannot be evaluated. In this region, resistance separation was conducted using the correlation between EIS and the polarization curve.

### 4.2. Comparison of the Effective Protonic Resistance in CCL and Membrane Resistance with RH and Current Density

As the electronic bulk and contact resistances in *HFR* remained constant with variations in RH and current density, it was inferred that the change in *HFR* was caused by the variation in membrane resistance. As shown in Figure 7a,b, $R_{ohmic}$ was divided into *HFR* and $R_P^{eff}$ using EIS analysis and polarization curve in the experiment in $H_2$/air. It is already well known that the protonic resistance in Nafion is mostly determined by the water content [40,41]. In addition, the water content is closely related to RH and current density. As RH and current density were varied, $R_{ohmic}$ notably changed owing to $R_P^{eff}$, but *HFR* was affected only a little. In other words, compared with *HFR*, $R_P^{eff}$ was more sensitive to the operating current density and RH condition because of the water content. Furthermore, the change in $R_P^{eff}$ was also highly related to the mean distance of ionic transport. The reaction rate and proton flow increased with current density. This means that in the low current density region, the effect of the increase in water content owing to the reaction was more critical than that of the increase in proton flow. This effect was equilibrated around the middle current density region, and thus, $R_P^{eff}$ became constant. Thereafter, $R_P^{eff}$ increased again owing to the high proton flow in the high current density region.

Additionally, the lowest levels of $R_P^{eff}$ were almost the same at identical current densities (1.3–1.8 A cm$^{-2}$) regardless of the RH condition. It appeared that the CCL was fully hydrated at these operating conditions, although the RH was 80%.

The summation of the electronic bulk resistance and contact resistance in *HFR* was 0.026 $\Omega$cm$^2$. This is a reasonable value when compared with that obtained in a previous study [13]. Based on this measurement, the protonic resistance in the membrane can be

evaluated. Therefore, the protonic resistance in the membrane was greater than in the CCL in the middle and high current density regions shown in Figure 7a,b. One of the reasons for this is the thickness of the MEA, which was measured using scanning electron microscope (SEM) images. The thickness of the membrane was $15.00 \pm 1.5$ µm, and that of the CCL was $10.91 \pm 1.5$ µm. Based on Ohm's law, the resistance in the membrane should be greater than that in the CCL, similar to the trend observed in the experimental results. The second reason pertains to the diffusion effect and distance of proton flow. The protons migrated through the electrolyte, but the distance of movement changed with the diffusion of the reactant gas and proton conductivity. In the membrane, the protons migrated from the anode to the cathode; thus, the distance of proton movement was greater than or equal to the thickness of the membrane. However, in the CCL, the distance of proton movement was less than or equal to the thickness of the CCL because $O_2$ also migrated to the triple phase boundary. Hence, the effective diffusion coefficient ($D^{eff}$) affected the $R_P^{eff}$. To confirm this effect, a $H_2$/Heliox experiment was conducted to vary $D^{eff}$, as shown in Figure 7c. As a result, $R_P^{eff}$ decreased at high current densities, because the distance of proton flow in the CCL decreased. In other words, several reactant points were moved to the nearby membrane.

$R_P^{eff} \approx R_p/3$ is a well-known equation used to evaluate $R_P^{eff}$. This equation can be derived from Equation (16), and the error can be estimated as well. To derive the equation, $r_p/(r_{ct} + r_{mass})$ was assumed to be very small ($r_p/(r_{ct} + r_{mass}) \approx 0$). This was stricter than the assumptions with Equation (8) to (13). Therefore, the error owing to the assumptions with Equation (8) to (13) was less than 1.5%, but the error due to Equation (23) was more than 5.8% at RH 80% and 7.1% at RH 100%. In this context, $R_P^{eff}$ was roughly $R_p/3$, but this assumption was not validated in the main operating load range, as shown in Figure 7.

## 5. Conclusions

The separation of resistances was investigated using EIS analysis and polarization curves. To analyze the EIS, a general solution based on a recursion formula was derived and validated. The solution was simplified using three assumptions, and thus, $R_{total}$ was divided into $R_{ct}$, $HFR$, $R_P^{eff}$, and $R_{mass}$. When the approximation of $r_p/(r_{ct} + r_{mass})$ or the assumption of the homogeneity of $r_{ct}$ was not valid, the resistance separation was carried out using both EIS analysis and polarization curves. The $HFR$ was estimated from the EIS. The $R_{ct}$ and $R_{mass}$ values were estimated using a polarization curve with the semi-empirical equation. In this case, the equation was identified by fitting the EIS data in the other current density regions that satisfied the assumptions. Therefore, $R_P^{eff}$ was calculated using other resistances obtained from the polarization curve and EIS.

The experimental results of $R_{ohmic}$ suggested that $HFR$ and $R_P^{eff}$ were sensitive to water content. Consequently, compared with $HFR$, $R_P^{eff}$ was more strongly dependent on the operating current density and the RH condition owing to the water content. In addition, $R_P^{eff}$ was less than $HFR$ because of the lower material thickness and change in the mean distance of ionic transport. To identify the effect of the movement differential, $H_2$/Heliox experiments were conducted. As a consequence, the $R_P^{eff}$ and $R_{mass}$ were varied owing to the diffusion effect. The $R_P^{eff}$ was changed in the $H_2$/Heliox experiment, especially at high current densities, because the diffusion coefficient was low under $H_2$/air.

The relationship $R_P^{eff} \approx R_p/3$ was additionally derived from the general solution, but this approximation entailed a relatively large error of more than 5.8% at RH 80% and 7.1% at RH 100% in this experimental condition.

The separation of resistances is helpful in the diagnosis of PEM fuel cells, and the calculation of overpotential is useful in identifying the effect of resistance in terms of voltage.

**Author Contributions:** Conceptualization and Methodology, J.C. and K.M.; Investigation, J.C. and J.S.; Analysis, J.C., J.S., H.O., and K.M.; Writing, J.C., J.S., H.O., and K.M.; Supervision, K.M. All authors have read and agreed to the published version of the manuscript.

**Funding:** This research received no external funding.

**Acknowledgments:** This work was granted financial resources from the SNU Institute of Advanced Machines and Design (IAMD).

**Conflicts of Interest:** The authors declare no conflict of interest.

## Abbreviations

| | |
|---|---|
| $r_p$ | distributed protonic resistance in the CCL, $\Omega$ cm |
| $r_{ct}$ | distributed charge transfer resistance, $\Omega$ cm |
| $z_W$ | distributed BCPE, $\Omega$ cm |
| $r_{mass}$ | distributed mass transport resistance, $\Omega$ cm |
| $q_C$ | parameter related to CPE, $\Omega^{-1}$ cm$^{-3}$ s$^{P_C}$ |
| $P_C$ | CPE exponent |
| $q_B$ | distributed parameter related to BCPE, $\Omega^{-1}$ cm$^{-3}$ s$^{P_B}$ |
| $P_B$ | BCPE exponent |
| $k$ | iteration number of the node, $1 \leq k \leq n-1$ |
| $n$ | total repeating number of the node |
| $HFR$ | high frequency resistance, $\Omega$ cm$^2$ |
| $R_p$ | protonic resistance in the CCL, $\Omega$ cm$^2$ |
| $R_p^{eff}$ | effective protonic resistance in the CCL, $\Omega$ cm$^2$ |
| $R_{ct}$ | charge transfer resistance, $\Omega$ cm$^2$ |
| $Z_W$ | BCPE, $\Omega$ cm$^2$ |
| $R_{mass}$ | mass transfer resistance, $\Omega$ cm$^2$ |
| $R_{ohmic}$ | ohmic resistance, $\Omega$ cm$^2$ |
| $R_{total}$ | total resistance, $\Omega$ cm$^2$ |
| $Q_C$ | parameter related to CPE, $\Omega^{-1}$ cm$^{-2}$ s$^{P_C}$ |
| $Q_B$ | parameter related to BCPE, $\Omega^{-1}$ cm$^{-2}$ s$^{P_B}$ |
| $D^{eff}$ | effective diffusion coefficient of oxygen diffusion in the CCL, cm$^2$ s$^{-1}$ |
| $x$ | non-dimensional distance along the catalyst layer, $0 \leq x \leq 1$ |
| $l$ | thickness of the CCL, cm |
| $\sigma$ | proton conductivity, $\sigma = \sigma_{el}/l$, S cm$^{-2}$ |
| $\sigma_{el}$ | specific proton conductivity, S cm$^{-1}$ |
| $m_{mass}$ | mass transport coefficient, V |
| $n_{mass}$ | simulation parameter for the polarization curve fitting, cm$^2$ A$^{-1}$ |
| $b$ | Tafel slope, V |
| $E$ | cell voltage, V |
| $E_0$ | open-circuit voltage (OCV), V |
| $j$ | current density distribution along the CCL, A cm$^{-2}$ |
| $i^*$ | exchange current density, A cm$^{-2}$ |
| $I$ | current density, A cm$^{-2}$ |
| $\eta_1$ | overpotential at the PEM/CCL interface, V |
| $i$ | imaginary component in impedance |
| $w$ | frequency, Hz |
| $F$ | Faradaic constant, $96,487$ A s mol$^{-1}$ |
| $P$ | total pressure, Pa |
| T | temperature, K |
| R | Gas constant, $8.314$ J K$^{-1}$ mol$^{-1}$ |

**Subscripts**

| | |
|---|---|
| *C* | constant phase element (CPE) |
| *B* | bounded constant phase element (BCPE) |
| *ct* | Charge transfer |
| *ohmic* | ohmic |
| *mass* | mass transport |
| *k* | iteration number of the node, $1 \leq k \leq n-1$ |
| *n* | total repeating number of the node |

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
