# Peer review of "Resistance Separation of Polymer Electrolyte Membrane Fuel Cell by Polarization Curve and Electrochemical Impedance Spectroscopy"

_energies, doi:10.3390/en14051491_

Round 1
Reviewer 1 Report
The tests were performed using a commercial Gore MEA. It would be useful if the article included an assessment of how specific the results are to the MEA being used.
Author Response
We thank Reviewer #1 for this assessment and have revised the manuscript substantially to address these failings of the original manuscript. We believe that we have sufficiently addressed the concerns of Reviewer#1 to allow consideration of publication in the Energies
- A commercialized MEA was used to improve the validity and reproducibility of the experimental results.
Changes : revision in page 3 line 6
*Image files also in the attachment

Reviewer 2 Report
Dear authors,
I have read your manuscript entitled "Resistance Separation of Polymer Electrolyte Membrane Fuel Cell by Polarization Curve and Electrochemical Impedance Spectroscopy" and I have found it well written, interesting and sound. Nevertheless, prior consideration of publication in its final form, I would like you to suggest some considerations:
- High frequency region is cited in the Introduction but the frequency range you are referring is not provided, please clarify the range.
- Fig 2.b. is difficult to read in its present form. Please consider increasing its size (you have space enough in the page) to make it more readable.
- Fig 7. Bold letters inside this figure are not necessay. Titles of the series would be clearer in normal typping.
- Acronym SEM appears for the first time in page 14 line 339 but it has not been defined yet. Please define.
Modifying these aspects, will improve your manuscript considerably. Additionally, your analysis, explanation, and research regarding the use of EIS and Polarization Curves has been excelent. Congratulations to the authors.
Regards,
Author Response
We thank Reviewer #2 for this assessment and have revised the manuscript substantially to address these failings of the original manuscript. We believe that we have sufficiently addressed the concerns of Reviewer#1 to allow consideration of publication in the Energies
- For the sinusoidal alternating current (AC) signal, the frequency range analyzed was 20 kHz to 900 mHz. The high frequency resistance was measured at 20kHz resistance.
Changes : revision in page 3 line 15 - We increased the resolution and size of the Figure 2.(b)
- The Fig. 7 bold letters were changed to normal letter.
- The definition of acronym SEM was edited.
*Image files of the graphs or screenshot are in the attachment

Reviewer 3 Report
This paper studies the separation of resistances using EIS analysis and polarization curves. To analyze the EIS, a general solution based on a recursion formula is proposed and validated by simplifying the solution using three assumptions. The resistances were separated and analyzed with respected to variations in relative humidity in the entire current density region. In the case of ohmic resistance, high frequency resistance was almost constant in the main- operating load range (0.038–0.050Ω cm2), while protonic resistance in the CCL exhibited sensitivity owing to oxygen diffusion and water content.
The study deserves to be published in Energies after the following improvements:
- The term “The separation of resistances” in abstract should be changed in “The separation of resistances during their measurement…” or something indicating that the separation is intended in the measurement.
- In the abstract and introduction section: it could be clarified that the novelty of the paper is the use of EIS technique to separate ionic from electronic and activation impedance inside the catalyst layer (If I have understood correctly). However,
- Line 91: with the term “total resistance” is intended the resistance of the full cell (anoce/electrolyte/cathode/current collectors)? Please clarify.
- More words (not only commercial names) specifying the chemical nature of the materials composing the fuel cell should be provided.
- Line 154: correct typos
- Please add quantitative information on reproducibility of the experimental results.
Author Response
We thank Reviewer #3 for this assessment and have revised the manuscript substantially to address these failings of the original manuscript. We believe that we have sufficiently addressed the concerns of Reviewer#1 to allow consideration of publication in the Energies
- ‘The separation of resistances’ was changed in ‘The separation of resistances during their measurement’.
- ‘To separate ionic resistance from electronic and activation impedance inside the catalyst layer’ is the novelty of the paper. We can clarify this one using equations.
Please see the attachment to check the equations.
( "210304-EnergiesReview_Reply_Reviewer3" ) - We tried to find the typo throughout page 5~6 (Line154), but we could not find it.
